# Yield of genetic association signals from genomes, exomes and imputation in the UK Biobank

Sheila M. Gaynor [1,3] ✉, Tyler Joseph[1,3], Xiaodong Bai[1], Yuxin Zou[1], Boris Boutkov[1], Evan K. Maxwell [1], Olivier Delaneau [1], Robin J. Hofmeister[2], Olga Krasheninina[1], Suganthi Balasubramanian [1], Anthony Marcketta[1], Joshua Backman [1], Regeneron Genetics Center*, Jeffrey G. Reid [1], John D. Overton[1], Luca A. Lotta [1], Jonathan Marchini [1], William J. Salerno [1], Aris Baras [1] ✉, Goncalo R. Abecasis [1,4] ✉ & Timothy A. Thornton [1,4] ✉

Whole-genome sequencing (WGS), whole-exome sequencing (WES) and array genotyping with imputation (IMP) are common strategies for assessing genetic variation and its association with medically relevant phenotypes. To date, there has been no systematic empirical assessment of the yield of these approaches when applied to hundreds of thousands of samples to enable the discovery of complex trait genetic signals. Using data for 100 complex traits from 149,195 individuals in the UK Biobank, we systematically compare the relative yield of these strategies in genetic association studies. We find that WGS and WES combined with arrays and imputation (WES + IMP) have the largest association yield. Although WGS results in an approximately fivefold increase in the total number of assayed variants over WES + IMP, the number of detected signals differed by only 1% for both single-variant and gene-based association analyses. Given that WES + IMP typically results in savings of lab and computational time and resources expended per sample, we evaluate the potential benefits of applying WES + IMP to larger samples. When we extend our WES + IMP analyses to 468,169 UK Biobank individuals, we observe an approximately fourfold increase in association signals with the threefold increase in sample size. We conclude that prioritizing WES + IMP and large sample sizes rather than contemporary short-read WGS alternatives will maximize the number of discoveries in genetic association studies.

Large-scale genetic studies provide insight into the biological underpinnings of a wide range of human traits, guiding the development of new therapeutic interventions and disease prevention strategies through improved understanding of human health and disease. Examples of new therapies enabled by genetic studies include blocking *PCSK9* for the prevention of recurrent heart disease[1–3], blocking *ANGPTL3* for the treatment of familial hypercholesterolemia[4] and CRISPR editing of *BCL11A* for the treatment of sickle cell disease[5,6]. Genetic association studies can now include extensive health data across millions of participants and have a choice of diverse approaches for capturing genetic variation, ranging from array genotyping with imputation (IMP) to WES to WGS. Each of these approaches differs in the number and type of variants captured, the fidelity of information provided and the cost and complexity of execution. However, the impact of choosing each approach on the detection of actionable genetic association signals remains unclear.

A full list of affiliations appears at the end of the paper. *A list of authors and their affiliations appears at the end of the paper.
✉e-mail: sheila.gaynor@regeneron.com; aris.baras@regeneron.com; goncalo.abecasis@regeneron.com; timothy.thornton@regeneron.com

For the past 15 years, genotyping and imputation has been the workhorse of genetic association studies[7,8]. Studies that use array genotyping followed by continually improved imputation are routinely applied to complex traits ranging from macular degeneration[8] to inflammatory bowel disease[9,10] to schizophrenia[11] and have produced tens of thousands of genetic association signals[12]. IMP enables the study of relatively common variants (typically, those with frequencies of >0.1–1% that have been characterized in a reference panel of individuals)[13]. Recent examples include studies of COVID-19 susceptibility in hundreds of thousands of individuals, identifying genetic variants that lower *ACE2* expression as protective factors and identifying variants in *IFNAR2* and other immune-related genes as major determinants of susceptibility[14,15]. Nonetheless, translating genome-wide association study (GWAS) findings into actionable insights for biological understanding and therapeutic intervention remains a laborious and ongoing process[16]. Challenges in translation arise because most GWAS findings point to non-coding variants, variants whose function is uncertain and/or variants with small effect sizes.

In the past 10 years, exome sequencing has emerged as a practical and successful strategy for uncovering the genetic basis of human disease. WES captures protein-coding variants as rare as singletons that are beyond the reach of arrays and IMP. Exome sequencing was originally used to identify the causes of hundreds of rare Mendelian disorders by studying collections of individual cases and their families[17]. Exome sequencing studies of complex traits in large population-based samples are increasingly common. They have yielded rare coding variant association signals that are easier to interpret and experimentally follow up than those found in GWAS. Translation for these signals is easier when variants have large effect sizes and connect the impaired function of a specific gene to a therapeutic outcome. Recent examples include studies of rare genetic variants that protect against obesity and liver disease[18,19], pointing to *GPR75* and *CIDEB*, respectively, as potential therapeutic targets. WES and IMP are often used together as an effective approach for capturing both common variants genome-wide and rare protein-altering variants in coding regions.

Most recently, short-read WGS has been applied at scale to dissect Mendelian diseases[20] and common diseases[21–23]. WGS studies aim to capture coding and non-coding variation across the allele frequency spectrum and interrogate a much larger collection of genetic variants than either IMP or WES-based approaches. There is widespread excitement about the potential of WGS for enabling genetic discovery[24,25]. WGS has the most substantial resource and time costs, including data generation, processing, storage and analysis, but these are decreasing[26]. At present, it remains unknown whether the deployment of WGS at scale will enable a wave of genetic discoveries comparable to those that resulted from the deployment of arrays, IMP and WES.

Understanding the relative performance of these approaches is essential for the design of current and future genetic studies. Here, we systematically assess variants captured and the resulting discovery yield for genetic association studies in large biobank samples using IMP, WES and WGS. For these comparisons, we use data from 149,195 UK Biobank (UKB) participants who have been characterized with each of these assays and for whom extensive health information is available. We first perform a survey of the genetic variants that can be assessed with each approach and then compare the yield of single-variant and gene-based association signals across a set of 100 traits in this large UKB sample. Finally, recognizing that the choice of analysis approach could entail changes in sample size, we evaluate association yield when varying the sample size using the same set of 100 traits. Our study provides an empirical assessment of these approaches to facilitate investigators in making informed decisions about large-scale genetic studies.

## Results

### UKB dataset

Our primary analyses included 149,195 UKB participants with data from IMP[27], WES[28,29] and WGS[21] as described in Supplementary Figs. 1 and 2 and Supplementary Table 1. Most of the analytical sample was obtained from individuals of European ancestry (EUR; 95%), with the remaining individuals having African (AFR; 2%), South Asian (SAS; 2%) and other ancestries (<1%). The UKB used a custom-designed single-nucleotide polymorphism (SNP) array with 805,426 variants designed to capture common variations and selected high-value protein-altering variants. We extended these data to 111,333,957 variants through imputation using TOPMed Freeze 8 genomes as a reference panel, which is an ancestrally diverse WGS panel comprising genomes of EUR, AFR, Hispanic and Latino (HLA), Asian and other or multiple ancestries (see Supplementary Fig. 12 for an evaluation of imputation performance by ancestry)[22,30,31]. WES targeted the coding region of 18,893 genes (~1% of the genome) and sequenced 95% of these bases to a depth of >25× in each individual[28,29]. WES resulted in 17,131,8674 variants, of which 10,463,945 were in coding regions. Finally, WGS was carried out to a depth of >20× using short-read sequencing, ensuring the capture of >95% of the genome with >15× depth. Overall, WGS resulted in 599,385,545 variants[21]. We applied uniform processing and filtering to the variant sets from each approach (see Methods for details; key filtering steps included removing variants known to have failed in prior large-scale WGS efforts). We evaluated four different strategies: IMP, WES, WGS and WES + IMP; full results for each strategy are included in tables, figures and supplementary information; however, for ease of presentation, we focus on the comparison of WES + IMP and WGS, which were the two approaches with the highest yield of genetic association signals.

Overall, the WGS strategy identified 599,385,545 single-nucleotide variants and indels, whereas the WES + IMP strategy identified 125,694,205 single-nucleotide variants and indels (Fig. 1a,b). Variants jointly captured by each of the platforms were 99.9% concordant (Supplementary Table 2). In contrast to the total number of variants, the count of variants observed in each individual was similar between the two approaches, with a mean of 3,595,704 variant alleles per individual in WGS and 3,585,289 in WES + IMP (Fig. 1c). The bulk of variants unique to WGS are very rare and present in only a few individuals each, explaining how an approximately fivefold increase in overall variant count translates into only ~0.3% more variant alleles per individual (Fig. 1d). For example, 47% of WGS variants are singleton variants present in one individual, whereas only 7% of WES + IMP variants are singletons.

Variant identification in coding regions was very similar: the total number of coding variants was 6,732,108 variants for WGS versus 6,761,880 variants for WES + IMP, and the mean count of coding variant alleles per individual was 20,000 for WGS versus 20,039 for WES + IMP (Fig. 1c). Within coding regions, 48% of variants were singletons for both WGS and WES + IMP. The differential contributions of WES and IMP to the coding and non-coding variant sets in WES + IMP are described in Fig. 1e. Coding variation is further described in the Supplementary Note, Supplementary Fig. 3 and Supplementary Tables 3 and 4.

### Single-variant tests

We assessed differences in genetic association yield in two stages. First, we evaluated differences in yield for single-variant tests (which are responsible for the bulk of known genetic association signals). Second, we evaluated differences in yield in gene-based tests (discussed in the next section; Supplementary Fig. 4), which can provide more specific insight into the underlying biology. We performed association tests across 80 quantitative and 20 binary traits and first tested each variant present in at least five individuals for association using REGENIE[32] (see Methods for further details). We used a significance threshold of $P = 5 \times 10^{-12}$ for the main analysis; a threshold of $P = 5 \times 10^{-11}$ was used for classifying a signal as shared across platforms when at least one

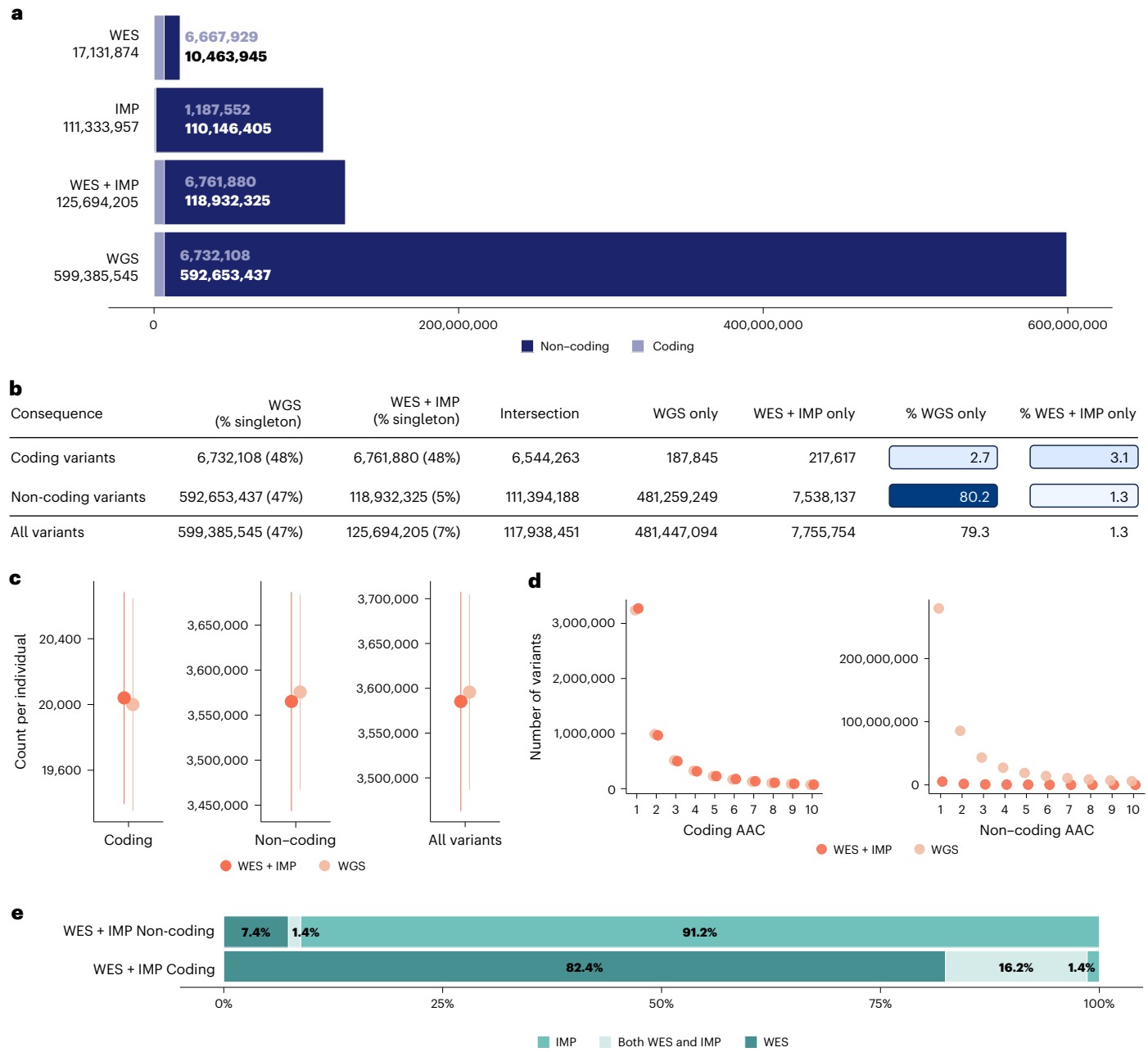

**Fig. 1 | Genetic variation captured by genomes, exomes and array genotyping with imputation. a**, The number of coding and non-coding variants for all variants in the WES, IMP, WES + IMP and WGS datasets. **b**, A comparison of variants observed by the WES + IMP and WGS datasets by functional consequence, with the relative gains in approach-exclusive variants. **c**, The number of variant alleles observed per individual (*n* = 149,195).

The point provides the sample mean; error bars, ±standard error of the mean (s.e.m.) **d**, The number of non-coding and coding variants observed at the lowest alternative allele counts (AAC) for the WES + IMP and WGS data. **e**, The percentages of non-coding and coding variants observed only in WES, in both WES and IMP and only in IMP in the combined WES + IMP dataset.

platform reached $P = 5 \times 10^{-12}$. The association results for all tests are summarized in Fig. 2.

We identified 3,570 genome-wide significant signals across all trait–locus pair combinations (Supplementary Data 1). We found a similar number of signals from WES + IMP (3,506 or ~35.1 per trait) and from WGS (3,534 or ~35.3 per trait, a 1% increase). Nearly all signals were found using both approaches (3,470 of 3,570, or 97.2%), and most signals pointed to the exact same variant (96%). Among the 4% of signals that pointed to different variants, we found effect sizes to be similar (Pearson's *r* = 0.86). Supplementary Fig. 5 compares frequencies, effect sizes and *P* values across signals; Supplementary Table 5 summarizes the annotated functional consequence of the peak variant for each

signal. Across all platforms, the same highly significant associations with the largest $-\log_{10} P$ value typically corresponded to high allele frequency variants and small effect sizes. Shared signals were comparable beyond the peak variant (Supplementary Fig. 6 summarizes a typical shared signal across platforms). Such shared associations included signals in genes used to inform therapeutics; for instance, the peak signal in *PCSK9* with low-density lipoprotein cholesterol and missense variant 1:55039974:G:T (alternate allele frequency, 0.018) was identified across WGS ($P = 4.18 \times 10^{-140}$), WES ($P = 4.18 \times 10^{-140}$) and IMP ($P = 6.23 \times 10^{-140}$).

In contrast to the shared signals, nearly all signals that were exclusive to either WGS (64 signals) or WES + IMP (36 signals) were relatively

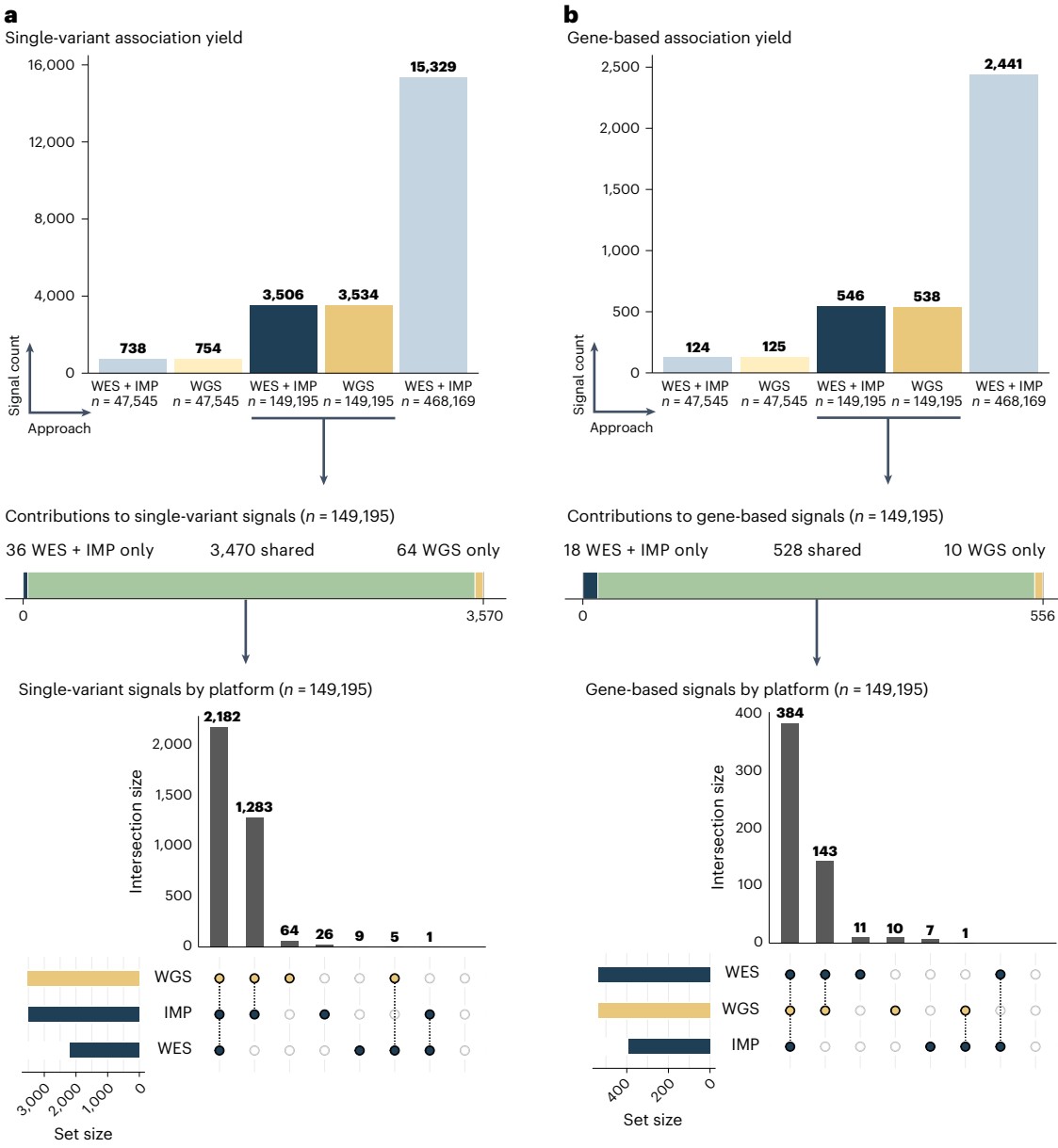

**Fig. 2 | Single-variant and gene-based association signals from genomes, exomes and array genotyping with imputation. a**, Single-variant association signals identified across each analysis sample, where platform-specific results are specified for the primary analytical sample ($n = 149,195$). Summarized results are also provided for the full UKB sample with WES + IMP data ($n = 468,169$) and a subset of the UKB sample ($n = 47,545$), downsampled to also have a 1:3 ratio with the primary analytical sample used for the majority of our analyses ($n = 149,195$). **b**, Gene-based association signals identified across each analysis sample, where platform-specific results are specified for the primary analytical sample ($n = 149,195$).

marginal in $P$ value (60% within two orders of magnitude of the significance threshold), pointed to rare variants with a frequency of <0.1% (67%) and typically had limited support from nearby variants. Supplementary Figs. 7–9 show examples of signals specific to each platform. We sought to replicate our findings using WES + IMP data from the remaining UKB individuals ($n = 318,974$). At the genome-wide significance level, we replicated 17 out of the 36 signals only in WES + IMP, 13 out of the 64 signals only in WGS and 3,386 out of the 3,570 shared signals. We interpret the observation that platform-specific signals are often relatively weak, led by rare variants, lack support from nearby variants and failed to replicate as evidence of platform-specific artifacts for many of these signals.

We also performed the association analysis described above using a less conservative significance threshold of $P = 5 \times 10^{-10}$, which resulted in more genome-wide significant signals, as expected, but did not change the overall conclusions on association yield for the different approaches (Supplementary Table 7).

## Gene-based tests

In contemporary genetic association studies, gene-based tests—which aggregate evidence across rare protein-altering variants in a gene—provide a powerful complement to single-variant tests and are especially important for identifying biological insights. These tests directly point to biological effector mechanisms and can often identify sets of variants with very large effects[28,29]. Therefore, we followed up our analysis of single variants with a series of gene-based tests. We examined coding variants in each gene grouped according to frequency (alternate allele frequencies of <1%, <0.1%, <0.01%, <0.001% and singletons) and functional consequence (missense, deleterious missense and predicted loss of function (pLoF)) and considered association tests for which

all variants in a gene act in the same or different directions (Methods and Supplementary Table 6). We summarized all association tests for gene−trait pairs using GENE_P[33], a single, unified gene-based $P$ value in which a significance and replication threshold of $P = 2.6 \times 10^{-8}$ was used (Methods). We found the overall conclusions from the results to be similar when examining the individual gene-based tests using different allele frequency thresholds or applying a less conservative significance threshold (Supplementary Table 7).

We observed significant associations between 556 gene−trait pairs across the WES + IMP and WGS analysis (Supplementary Data 2). We again found a similar number of signals from WES + IMP (546) and from WGS (538, a 1% decrease). Consistent with the single-variant analysis, 95% of gene-based association signals (528 out of 556 unique signals) were identified by both WES + IMP and WGS. Again, associations with the largest $-\log_{10}(P)$ value were consistently observed in WGS and WES + IMP (Supplementary Figs. 10 and 11). The shared signals included several known true positives, including therapeutically relevant findings such as the *PCSK9* association with low-density lipoprotein cholesterol (WGS, $P = 1.27 \times 10^{-80}$; WES, $P = 3.91 \times 10^{-82}$; IMP, $P = 1.42 \times 10^{-36}$; Supplementary Data 2).

When we examined the ten signals exclusive to WGS and the 18 signals exclusive to WES + IMP, we found that half were signals for which one approach exhibited sub-threshold association ($2.6 \times 10^{-7} < P < 1 \times 10^{-5}$), and the addition of a small number of platform-specific variants increased signals to significance for the alternate approach. The remaining cases were driven by a single variant that was platform-exclusive (one variant in *CALR* for WGS, one variant in *OMA1* in WES, six different variants in IMP). We again sought to replicate these signals in the remaining UKB samples and replicated 14 out of 18 signals exclusive to WES + IMP, 5 out of 10 signals exclusive to WGS and 514 out of the 556 shared signals.

### The importance of sample size

Advancing our understanding of human health and disease requires not only balancing the potential returns from each technology when applied to the same samples but also considering the consequences of examining different numbers of samples. To assess the impact of sample size on association study yield, we repeated all analyses on a subset of $n = 47,545$ individuals in UKB with WGS and WES + IMP and on the entire $n = 468,169$ UKB sample with WES + IMP data (Supplementary Figs. 1 and 2 and Supplementary Table 1), which correspond to approximately threefold changes in sample size relative to our primary analysis with $n = 149,195$ individuals.

When we focused our analyses on the set of 47,545 individuals (a threefold decrease in sample size), the number of single-variant signals decreased more than fourfold, from 3,534 to 754 for WGS and from 3,506 to 738 for WES + IMP, and the number of gene-based signals decreased more than fourfold, from 538 to only 125 for WGS and from 546 to 124 for WES + IMP. When we extended the WES + IMP analysis from 149,195 individuals to 468,169 individuals (a threefold increase in sample size), the number of single-variant signals increased more than fourfold, from 3,506 to 15,329. Similarly, the number of gene-based association signals also increased fourfold, from 546 to 2,441. Larger sample sizes permitted the identification of additional rare variant signals, often for variants present in the smaller dataset but with insufficient power. Comparing the analyses of 47,545 individuals to the analyses of 468,169 individuals, we find that the approximately tenfold increase in sample size resulted in an approximately 20-fold increase in association yield.

Choosing between WGS and WES + IMP had a limited impact on genetic association yield (changing the number of signals by about 1–2%) in both the subset sample ($n = 47,545$) and the main analysis sample ($n = 149,195$). However, increasing the sample size had an outsized increase on overall yield; thus, changes in experimental design that enable larger samples through lab efficiencies, cost and analytical simplicity can have a much greater impact on discovery yield than choices between these approaches. For example, WES + IMP for all UKB samples has been publicly available since November 2021, whereas the complete WGS data only became available in November 2023.

### Signals from each approach alone

Our analyses thus far have shown that the combination of WES + IMP is equivalent to WGS for practical purposes when focused on association yield. It can also be important to consider the relative yield from WES alone and IMP alone.

For single-variant association signals, we found IMP and WGS to be roughly equivalent. Nearly all the variants that could drive a significant association signal by themselves were common enough to be imputed. Even though 90% of peak variants were non-coding (Supplementary Table 5), WES alone was able to identify 62% of single-variant association signals. WES was able to find these signals because common variant association signals are enriched near genes and because, through linkage disequilibrium, most single-variant association signals are supported by nearby variants. However, using WES to detect non-coding single-variant signals resulted in a substantial loss of fidelity, often identifying different peak variants than WGS and IMP (which agreed 97% of the time).

Broadly speaking, WES and WGS found the same gene-based association results, whereas IMP alone missed 29% of the gene-based signals. In particular, signals driven by very rare variants could not be identified by IMP alone. For example, there were 81 gene−trait pairs for which WGS and/or WES found an association between a burden of singleton coding variants (whether pLoF or missense) and a trait at $P < 2.6 \times 10^{-8}$. None of those singleton signals could be found through IMP alone, although 38% could be identified by IMP through burden tests that grouped more common variants. In terms of mechanistic insight, these singleton signals are among the most compelling: they refer to loci for which a group of individuals, each with a unique defect in the same gene, associates with an altered phenotype. Signals driven by rare coding variation also corresponded to some of the largest effect association signals. While WGS and WES found quantitative trait signals associated with a change in trait values of >1 standard deviation for 71 and 68 gene−trait pairs, respectively, IMP found only 15 such signals. For binary traits, there were eight gene-based signals associated with a twofold or more increase or decrease in disease risk for WES and WGS, but only two such signals for IMP. For example, in the *GCK* gene, rare pLoF and deleterious missense variants with a frequency of <0.0001 are strongly associated with type 2 diabetes in WES (odds ratio, 10.0; $P < 10^{-20}$) and WGS (odds ratio, 10.1; $P < 10^{-22}$) but not in IMP (all $P > 0.01$)[34–36].

## Discussion

Genetic association studies can elucidate human biology as well as support and guide the development of new, life-changing therapies that improve health[37]. Understanding the different strategies for maximizing the yield of these studies, particularly through disease-associated signals with clear mechanistic implications, is critical to enable the best use of limited resources. In this paper, we describe the results of a biobank-scale empirical assessment of genetic discovery yield from WGS, WES and IMP. Using data from 149,195 deeply characterized samples from the UKB, we conducted a head-to-head comparison of these approaches at scale. Although WGS increased the total number of assayed variants from 125,694,205 to 599,385,545, the total number of association signals from WGS differed by only 1% from WES + IMP (from 3,506 to 3,534 for single variants and 546 to 538 for gene-based tests). Some have argued that IMP with a well-matched reference panel may be sufficient for genetic association discovery, particularly in population isolates[38]. In UKB, we found that sequencing-based approaches enable discoveries at frequencies below those accessible through IMP, increasing gene-based findings by 30% so that a combination of WES + IMP was

most effective for UKB. We found no clear advantage of WGS over the strategy of combining WES and IMP (WES + IMP) for single-variant and gene-based association discovery when measured as association yield.

The limited additional association yield from WGS is probably because most variants specific to WGS are non-coding and very rare. Detecting signals with rare non-coding variants requires grouping them with variants of similar function, which presently cannot be done effectively and remains a major challenge. Although coding variants have been effectively grouped into aggregated sets to analyze the lowest-frequency variants, our ability to classify, group and interpret non-coding variants is not yet advanced enough to similarly analyze the rarest non-coding variants. In the future, if such aggregate approaches can successfully group rare non-coding variants, this could provide a potential benefit of WGS.

Each of the approaches we evaluated can be used to capture other types of variants in addition to SNPs and indels. For instance, we analyzed 637,039 structural variants (SVs) (median, 7,952 structural variants per individual) detected in the WGS data, but these structural variants resulted in only 16 additional association signals (3,586 with SVs and 3,570 without SVs; Supplementary Table 8) in addition to those captured from single-nucleotide variants and indels across the 100 traits examined. As sequencing approaches evolve beyond the current short-read standards, it may be possible for variants other than SNPs and indels to make larger contributions to discovery.

The IMP component of the WES + IMP approach requires a suitable WGS reference panel for imputation. There are still many populations that are insufficiently represented in contemporary reference panels, and there remains a need for diverse WGS reference datasets to increase imputation accuracy and genetic discovery across all ancestries. In our IMP analysis with the TOPMed WGS panel, we found that imputation accuracy in UKB varied by ancestry, with the highest accuracy for EUR and AFR ancestry individuals; by contrast, East Asian (EAS) and SAS ancestries had the lowest imputation accuracy (Supplementary Fig. 12). Alternative panels with improved performance in these ancestries and across the UK have been proposed (Supplementary Fig. 13 and ref. [39]). WGS studies that enable improved imputation reference panels[13,21,22,40] can thus enhance genetic association discovery for future studies.

The cost and complexity of biobank-scale WGS can be substantial compared to WES + IMP, yet our analysis indicated that the two approaches have a comparable association yield (1% higher yield for single variants and 1% lower yield for gene-based tests with WGS). We note that approaches for WGS and WES + IMP continue to evolve. For instance, sequencing labs at the Regeneron Genetics Center use a combined exome and common variant capture assay, eliminating the need for a separate genotyping array. Others have used low-coverage WGS in combination with WES and still others continue to develop long-read alternatives to extend the potential of WGS. Optimal use of resources must also consider the value of generating genetic data on additional samples versus obtaining deeper phenotypic characterization (perhaps proteomics and RNA sequencing) on the same samples as alternatives to broad WGS[41–44]. Selecting the best strategy for each study should also take into consideration study design and goals, considering, for instance, the cost of recruitment (which is often relatively modest for biobank studies but very high for studies of rare Mendelian disorders) relative to genome assay and analysis costs. Our results show that increasing sample size is an extremely effective strategy for improving discovery yield. For example, an approximately threefold increase in sample size resulted in an approximately fourfold increase in signals, and an approximately tenfold increase in sample size resulted in an approximately 20-fold increase in signals. Although we see greater discovery yield by increasing sample size, we importantly found that the limited difference in yield between WGS and WES + IMP was consistent across sample sizes.

Based on our large-scale analyses of IMP, WES and WGS, we find that allocating resources towards characterizing the largest possible sample sizes with WES + IMP should maximize genetic association discovery yield. Until WGS costs decrease further, it is our view that the greatest opportunity for WGS to further genetic association discovery is by increasing the diversity and availability of imputation resources, particularly for non-European ancestry populations. At the same time, we expect that as more targeted samples are sequenced with WGS, the performance of IMP-based approaches will continue to increase and thereby further decrease any potential relative advantage of biobank-scale WGS. Resource-efficient genetic studies supplemented with population-scale functional assays can empower discovery and facilitate the biological follow-up that is essential for translating genetic discoveries into benefits for human health.

## Online content

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

## Regeneron Genetics Center

**Sheila M. Gaynor[1,3], Tyler Joseph[1,3], Xiaodong Bai[1], Yuxin Zou[1], Boris Boutkov[1], Evan K. Maxwell[1], Olivier Delaneau[1], Olga Krasheninina[1], Suganthi Balasubramanian[1], Anthony Marcketta[1], Joshua Backman[1], Jeffrey G. Reid[1], John D. Overton[1], Luca A. Lotta[1], Jonathan Marchini[1], William J. Salerno[1], Aris Baras[1], Goncalo R. Abecasis[1,4] & Timothy A. Thornton[1,4]**

A list of members and their affiliations appears in the Supplementary Information.

## Methods

### Ethics statement

This study conducted analyses of phenotype and genetic data from the UKB cohort, under the approved UKB application license number 26041. The UKB project has ethical approval reviewed and provided by the North West Research Ethics Committee. Informed consent was provided by all study participants.

### UKB data preparation

The sample collection and preparation for the UKB has been previously described for each platform: arrays[27,45], WES[28,29] and WGS[21]. These approaches are summarized below.

**Array genotyping.** For array genotyping, DNA was extracted and provided in aliquots to Affymetric Research Services Laboratory for genotyping. Blood samples were genotyped using the UKB Axiom array with 805,426 variants. The samples were processed in 106 batches using a custom multi-batch genotype calling pipeline. Multiple quality checks including marker-based (evaluating new markers and effects based on factors such as batch, plate, sex and array) and sample-based (missing rates and heterozygosity) were performed.

**Exome sequencing.** For exome sequencing, DNA was prepared in fragments of 200 bp on average with 10 bp unique bar codes. Samples were processed with IDT's xGen probe library, PCR-amplified and quantified by quantitative PCR. Multiplexed samples were sequenced with 75 bp paired-end reads on the Illumina NovaSeq 6000 platform with S2 or S4 flow cells. The OQFE protocol was applied for reference alignment to GRCh38, variants were called using DeepVariant, aggregation was performed using GLnexus to generate a project-level pVCF file using an AQ1 cutoff of 20 and quality checks were performed. Using this approach, 96% of targeted bases were covered at a depth of >20×.

**Genome sequencing.** A pseudo-random subset of samples was selected for genome sequencing. DNA was prepared in fragments of 450–500 bp on average with barcode tracking. Samples were processed with IDT for Illumina, purified and pooled. Libraries were prepared from paired-end reads on the Illumina NovaSeq 6000 platform with S4 flow cells. The deCODE pipeline was applied for reference alignment to GRCh38, processing and merging, variant calling using GATK HaplotypeCaller and Graphtyper (v.2.7.1), and quality checks were performed. In this approach, samples had an average depth of >23.5×.

### Imputation of array datasets using the TOPMed Freeze 8 reference panel

Arrays were imputed using the TOPMed Freeze 8 reference panel on the Michigan Imputation Server[30]. Array variants were selected previously[29] based on UKB imputation use with HRC and ability to liftover to GRCh38 and uploaded in randomized batches for imputation on the server. The resulting VCF files were merged and concatenated, then subset to the individuals included in the present analysis. To retain high-quality imputed genotypes, variants with MAC > 5 and MACH $r^2$ > 0.3 were retained.

### Ancestry assignment

The continental ancestry of the sampled individuals was assigned as previously described[29]. In brief, principal components (PCs) were computed using the HapMap3 samples as reference, including all SNPs shared with the UKB array data, and then each of the UKB samples was projected onto the PCs. A kernel density estimator was trained and used to calculate the likelihood of a sample belonging to a continental ancestry group: AFR, HLA, EAS, EUR and SAS. Samples with a likelihood of a single ancestry greater than 0.3 were assigned that ancestry; samples with a likelihood of two ancestries greater than 0.3 were assigned AFR over EUR, HLA over EUR, HLA over EAS, SAS over EUR and HLA over AFR.

If ancestry likelihoods were all less than 0.3 or three ancestries were greater than 0.3, the sample was excluded from the analysis.

### Analysis-ready dataset preparation

**Sample selection.** To permit comparison across platforms, we identified all individuals for whom array genotyping, exome sequencing and genome sequencing were available. This yielded a sample of $n$ = 149,195 and was our primary analytical set. To evaluate the influence of sample size, we also constructed a sample from all individuals with both array genotyping and exome sequencing. This yielded a sample of $n$ = 468,169 individuals. We also generated a subset of individuals from our main analysis set by downsampling to retain the same multiplier (3.138) between sample sets. We randomly selected $n$ = 47,545 individuals with genome sequencing data for the final sample. Lastly, we generated a replication sample by selecting all $n$ = 318,974 individuals with WES + IMP that were not included in the primary analytical set of $n$ = 149,195 individuals.

**Genetic data.** Data were prepared for analysis within each platform using consistent filtering. For the sequencing datasets, all variants with MAC ≥ 1 were considered. The WES data were called using an AQ1 cutoff of 20. Across all genetic datasets, we excluded variants with Hardy–Weinberg equilibrium test $P$ > 1 × 10$^{-15}$ and >10% missingness. We further excluded all variants that failed quality control in TOPMed Freeze 8 (ref. 22) or failed quality control as given by the GraphTyper HQ definition in a previous publication[21]. Analysis-ready datasets were generated in Plink2 file sets (PGEN format).

**Phenotype data.** For the phenotype data, we used trait data from the UKB Data Showcase. A set of 100 traits (listed in Supplementary Data 3) was selected to permit generalizable conclusions on genetic association discovery yield for quantitative and binary traits. Traits were selected from 492 traits that were previously described and identified to have rare variant associations[29]. The trait set was reduced to retain 80 quantitative and 20 binary traits by pruning for redundancy, sufficient case counts and prioritizing trait heritability. Each of the quantitative traits was rank-based inverse-normal transformed before analysis.

### Gene and variant annotation

Variants were annotated using VEP[46] based on the canonical transcript of protein-coding transcripts. Gene regions were defined using Ensembl release 100, and canonical transcripts were defined using MANE tags where available, followed by APPRIS or Ensembl tags as necessitated. Variants were defined as pLoF when annotated as frameshift, stop gained, splice donor or acceptor, start lost, or stop lost. Missense variants were further assigned a deleteriousness score ranging from 0 to 5 based on five algorithms in dbNSPF[47]: SIFT[48], PolyPhen2 HDIV and HVAR[49], LRT[50] and MutationTaster[51]. Deleterious scores were grouped as 'likely deleterious' when predicted deleterious by five out of five algorithms, 'possibly deleterious' when predicted deleterious by at least one out of five algorithms and 'likely benign' when predicted deleterious by zero out of five algorithms.

### Association analyses

All association analyses were performed using REGENIE (v.3.1.1)[32]. For each of the 100 traits, we ran Step 1 of REGENIE using the observed genotyping array data. Array variants were included with <10% missingness and Hardy–Weinberg equilibrium test $P$ > 10$^{-15}$. The resulting predictors were included as covariates in Step 2 of REGENIE, in addition to age, age squared, sex, age-by-sex and ten ancestry-informative PCs derived from the array data. Our association analyses considered autosomes and chromosome X.

**Single-variant associations.** For single-variant analysis, each variant with MAC ≥ 5 was tested using Step 2 of REGENIE separately for each

platform. We considered a significance threshold of $P = 5 \times 10^{-12}$ based on Bonferroni correction for testing 100 phenotypes genome-wide. To identify independent signals, we performed peak-finding jointly across each dataset. This approach merged all sets of results (WGS, WES, imputed array) and identified independent peaks by scanning across the genome and identifying significant associations, then selecting the most significant signal and pruning signals to only permit one significant signal per 1,000 kb. To account for shared signals obscured by the significance threshold, we considered signals to be shared when a variant has a $P$ value within one order of magnitude of the significance threshold ($P = 5 \times 10^{-11}$) in the comparative platform; platform-specific unique signals did not have any signals within one order of magnitude of significance within 1,000 kb in the comparative platform. At each sample size, we excluded outlier significant associations from variants (112 associations) with high overall genotype mismatch (genotype discordance of >20%) identified in our main analysis sample from results when comparison was available across platforms. We evaluated outlier variants and excluded genotype mismatches again when considering a less conservative significance threshold as a result of the introduction of new signals. Replication was performed using the same approach for the replication sample (in WES, imputed array data), maintaining the significance threshold of $P = 5 \times 10^{-12}$.

**Gene-based associations.** We performed aggregation tests on rare variants using Step 2 of REGENIE separately for each platform to identify gene-based associations in coding regions. Annotations were used to generate gene sets by collapsing variants within gene regions based on allele frequency and functional consequence. We considered seven consequence-based masks (Supplementary Table 6): pLoF, pLoF + likely deleterious missense, pLoF + likely or possibly deleterious missense, pLoF + all missense, likely deleterious missense, likely or possibly deleterious missense, and all missense. We considered five allele frequency bins for each of the consequence-based masks, based on the alternative allele frequency thresholds: alternative allele frequency (AAF) ≤ 1%, AAF ≤ 0.5%, AAF ≤ 0.1%, AAF ≤ 0.01% and singletons only. Collectively, up to 35 mask combinations were tested for each gene when sufficient data were available for testing. A unified test was performed for each gene to yield a gene-level $P$ value that aggregated each mask combination across all testing frameworks (burden, SKAT and ACAT tests). We considered a significance threshold of $P = 2.6 \times 10^{-8}$ based on a Bonferroni correction for all gene-based tests within each platform; signals were defined as shared if the comparative platform had a $P$ value within one order of magnitude of significance ($2.6 \times 10^{-7}$). Platform-specific unique gene-based signals did not have $P < 2.6 \times 10^{-7}$ in the comparative platform. At each sample size, we excluded significant associations from sets with outlier variants (17 associations) with high heterozygous genotype mismatch (genotype discordance of >40%, a relaxed threshold compared to single-variant testing, as multiple variants contribute to gene-based signals) identified in our main analysis sample from results when comparison was available across platforms. We evaluated outlier variants and excluded genotype mismatches again when considering a less conservative significance threshold as a result of the introduction of new signals. Replication of significant associations was performed using the same approach with the replication sample (in WES, IMP data), maintaining the significance threshold of $P = 2.6 \times 10^{-8}$.

**Reporting summary**
Further information on research design is available in the Nature Portfolio Reporting Summary linked to this article.

**Data availability**
Full details of trait associations with variants and genes are available in Supplementary Data 1 and 2, respectively. UKB phenotype data, genotyping array data, WES data and WGS data can be accessed through the UKB research analysis platform (https://ukbiobank.dnanexus.com/landing). All data used in this research are publicly available to registered researchers through the UKB data-access protocol and who are listed as collaborators on UKB-approved access applications. The HapMap3 reference panel was downloaded from ftp://ftp.ncbi.nlm.nih.gov/hapmap. VCF files for TOPMED Freeze 8 were obtained from dbGaP as described in https://med.nhlbi.nih.gov/topmed-whole-genome-sequencing-methods-freeze-8. The human reference genome GRCh38 was obtained from http://ftp.1000genomes.ebi.ac.uk/vol1/ftp/technical/reference/GRCh38_reference_genome.

**Code availability**
All genetic association analyses were performed using REGENIE (https://rgcgithub.github.io/regenie). Data were prepared using GraphTyper (https://github.com/DecodeGenetics/graphtyper) and Genome Analysis Toolkit (v.4.0.12; https://gatk.broadinstitute.org/hc/en-us). Code to reproduce the analyses has been deposited at https://doi.org/10.5281/zenodo.13357248 (ref. 52).

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

**Acknowledgements**
This research was conducted using data from UKB, a major biomedical database. We thank all who made this work possible, including the UKB team, the member institutions who support this work and especially the UKB participants. The exome sequencing was funded by the UKB Exome Sequencing Consortium (Bristol Myers Squibb, Regeneron, Biogen, Takeda, Abbvie, Alnylam, AstraZeneca and Pfizer). The genome sequencing was funded by the UKB WGS Consortium (UK Research and Innovation, Wellcome Trust, Amgen, AstraZeneca, GlaxoSmithKline and Johnson & Johnson). Ethical approval for the UKB was previously obtained from the North West Centre for Research Ethics Committee (11/NW/0382); this work was approved by the UKB under application number 26041.

**Author contributions**
S.M.G., T.J., J.M., W.J.S., A.B., G.R.A. and T.A.T. conceived the study. A.M. and J.B. curated the data. X.B., O.K., W.J.S., J.G.R. and J.D.O. generated the data. S.M.G., T.J., X.B., Y.Z., O.D., R.J.H. and W.J.S. performed the formal analysis. J.D.O., J.G.R., A.B. and G.R.A. acquired funding. S.M.G., T.J., L.A.L., J.M., W.J.S., A.B., G.R.A. and T.A.T. were responsible for the methodology. B.B., E.K.M., S.B., J.G.R. and W.J.S. generated resources. J.D.O., L.A.L., J.M., W.J.S., A.B., G.R.A. and T.A.T. supervised the study. S.M.G., Y.Z. and R.J.H. visualized the study. S.M.G., T.J., G.R.A. and T.A.T. wrote the original draft of the paper; L.A.L., J.M., W.J.S. and A.B. contributed to writing and editing.

## Competing interests

S.M.G., T.J., X.B., Y.Z., B.B., E.K.M., O.D., O.K., S.B., A.M., J.B., J.G.R., J.D.O., L.A.L., J.M., W.J.S., A.B., G.R.A. and T.A.T. are current employees and/or stockholders of Regeneron Genetics Center or Regeneron Pharmaceuticals. The other authors declare no competing interests.

## Additional information

**Correspondence and requests for materials** should be addressed to Sheila M. Gaynor, Aris Baras, Goncalo R. Abecasis or Timothy A. Thornton.

# Reporting Summary

## Statistics

For all statistical analyses, confirm that the following items are present in the figure legend, table legend, main text, or Methods section.

| n/a | Confirmed | |
|---|---|---|
| ☐ | ☒ | The exact sample size (*n*) for each experimental group/condition, given as a discrete number and unit of measurement |
| ☐ | ☒ | A statement on whether measurements were taken from distinct samples or whether the same sample was measured repeatedly |
| ☐ | ☒ | The statistical test(s) used AND whether they are one- or two-sided<br>*Only common tests should be described solely by name; describe more complex techniques in the Methods section.* |
| ☐ | ☒ | A description of all covariates tested |
| ☐ | ☒ | A description of any assumptions or corrections, such as tests of normality and adjustment for multiple comparisons |
| ☐ | ☒ | A full description of the statistical parameters including central tendency (e.g. means) or other basic estimates (e.g. regression coefficient) AND variation (e.g. standard deviation) or associated estimates of uncertainty (e.g. confidence intervals) |
| ☐ | ☒ | For null hypothesis testing, the test statistic (e.g. *F*, *t*, *r*) with confidence intervals, effect sizes, degrees of freedom and *P* value noted<br>*Give P values as exact values whenever suitable.* |
| ☒ | ☐ | For Bayesian analysis, information on the choice of priors and Markov chain Monte Carlo settings |
| ☐ | ☒ | For hierarchical and complex designs, identification of the appropriate level for tests and full reporting of outcomes |
| ☐ | ☒ | Estimates of effect sizes (e.g. Cohen's *d*, Pearson's *r*), indicating how they were calculated |

*Our web collection on statistics for biologists contains articles on many of the points above.*

## Software and code

Policy information about availability of computer code

| Data collection | No software was used for data collection. |
|---|---|
| Data analysis | All genetic association analyses were performed using REGENIE (v3.1.1): https://rgcgithub.github.io/regenie/. Data was prepared using GraphTyper (v2.7.1): https://github.com/DecodeGenetics/graphtyper and Genome Analysis Toolkit (v.4.0.12): https://gatk.broadinstitute.org/hc/en-us. Code to reproduce analyses is available at GitHub (github.com/rgcgithub/ukb_genetic_association_yield). |

For manuscripts utilizing custom algorithms or software that are central to the research but not yet described in published literature, software must be made available to editors and reviewers. We strongly encourage code deposition in a community repository (e.g. GitHub). See the Nature Portfolio guidelines for submitting code & software for further information.

## Data

Policy information about availability of data

All manuscripts must include a data availability statement. This statement should provide the following information, where applicable:

- Accession codes, unique identifiers, or web links for publicly available datasets
- A description of any restrictions on data availability
- For clinical datasets or third party data, please ensure that the statement adheres to our policy

Full details of trait associations with variants and genes are available in Supplementary Data 1 and Supplementary Data 2, respectively. UKB phenotype data, genotyping array data, whole exome sequencing data, and whole genome sequencing data can be accessed via the UKB RAP: https://ukbiobank.dnanexus.com/

# Research involving human participants, their data, or biological material

Policy information about studies with human participants or human data. See also policy information about sex, gender (identity/presentation), and sexual orientation and race, ethnicity and racism.

| | |
|---|---|
| Reporting on sex and gender | Sex of study participants was collected as part of the UK Biobank study as described in Bycroft et al, Nature 2018 (https://www.nature.com/articles/s41586-018-0579-z).<br><br>54.2% of the participants are women.<br><br>Sex was used as a covariate in the regression models for genetic association testing. |
| Reporting on race, ethnicity, or other socially relevant groupings | Genetic ancestry and continental ancestry assignments for each participant were inferred from the genotyping array and exome sequencing data as described in Backman et al, Nature 2021 (https://www.nature.com/articles/s41586-021-04103-z). To control for confounding due to genetic ancestry differences among the UK Biobank participants, 10 ancestry-informative principal components (PCs) derived from the whole exome sequencing variants were obtained and included as covariates in the regression models for genetic association testing. |
| Population characteristics | The details of the population characteristics of the UK Biobank are described in the paper by Bycroft et al, Nature 2018 (https:// www.nature.com/articles/s41586-018-0579-z). Briefly, 94.7% of participants are of European ancestry, 54.2% are female, the average age at assessment is 58. Each participant reports 8 inpatient ICD10 3D codes, on average. |
| Recruitment | Please see Bycroft et al, Nature 2018. |
| Ethics oversight | Ethical approval for the UK Biobank was previously obtained from the North West Centre for Research Ethics Committee (11/NW/0382). Informed consent was provided by all study participants. The work described herein was conducted under the approved UK Biobank application number 26041. |

Note that full information on the approval of the study protocol must also be provided in the manuscript.

# Field-specific reporting

Please select the one below that is the best fit for your research. If you are not sure, read the appropriate sections before making your selection.

☒ Life sciences ☐ Behavioural & social sciences ☐ Ecological, evolutionary & environmental sciences

For a reference copy of the document with all sections, see nature.com/documents/nr-reporting-summary-flat.pdf

# Life sciences study design

All studies must disclose on these points even when the disclosure is negative.

| | |
|---|---|
| Sample size | Sample sizes were not predetermined. To permit comparison of genetic association yield across platforms, the intersection of samples with array genotyping, exome sequencing, and genome sequencing data available was used, which yielded a sample of n=149,195 individuals after quality control (QC) filtering. See Methods section "UKB data preparation" for details on how the QC was performed. To evaluate the influence of sample size on genetic association discovery yield we also compared the primary analytical sample to a sample of n=468,169 individuals with both array genotyping and exome sequencing data available. A randomly selected sample of n=47,545 individuals with genome sequencing data was also used to assess the impact of sample size, where the size of this down sample was chosen to retain the same multiplier (3.138) between the aforementioned two larger sample sets. Lastly, we generated a replication sample by selecting all n=318,974 individuals in the UK Biobank with both array genotyping and exome sequencing data available and who were not included in the primary analytical set of n=149,195 individuals. |
| Data exclusions | Variant level QC was performed as described in methods section "Analysis-ready dataset preparation" Genotyping array and sequencing data was prepared for analysis using consistent filtering. Across all platforms,, we excluded variants with Hardy-Weinberg equilibrium (HWE) test P > 1 x 10-15 and >10% missingness. We further excluded all variants that failed QC in TOPMed Freeze 8 or gnomAD, as well as additional QC filters as described in more detail in the manuscript.<br><br>A set of 100 traits were selected from 492 traits that were previously described in Backman et al, Nature 2021 (https://www.nature.com/articles/s41586-021-04103-z). A total of 80 quantitative and 20 binary traits were selected by pruning for phenotype redundancy, sufficient case counts, and prioritizing trait heritability. |
| Replication | A sample of n=318,974 individuals from the UK Biobank with both array genotyping and exome sequencing data available and that were not included in the primary analytical set of n=149,195 individuals was used for replication of genome-wide significant associations identified in |

the primary analytical dataset. We replicated (a) 17 of 26 single variant and 14 of 18 gene-based signals only from WES+IMP, (b) 13 of 64 single variant and 5 of 10 gene-based signals only in WGS, and (c) 3,386 of 3,570 shared single variant signals and 514 of 556 shared gene-based signals.

| Randomization | Individuals in the study were not being assigned to any experimental protocol or treatment and so randomization was not needed. |
| --- | --- |
| Blinding | Individuals in the study were not being assigned to any experimental protocol or treatment and so blinding was not needed. |

# Reporting for specific materials, systems and methods

We require information from authors about some types of materials, experimental systems and methods used in many studies. Here, indicate whether each material, system or method listed is relevant to your study. If you are not sure if a list item applies to your research, read the appropriate section before selecting a response.

## Materials & experimental systems

| n/a | Involved in the study |
| --- | --- |
| ☒ | ☐ Antibodies |
| ☒ | ☐ Eukaryotic cell lines |
| ☒ | ☐ Palaeontology and archaeology |
| ☒ | ☐ Animals and other organisms |
| ☒ | ☐ Clinical data |
| ☒ | ☐ Dual use research of concern |
| ☒ | ☐ Plants |

## Methods

| n/a | Involved in the study |
| --- | --- |
| ☒ | ☐ ChIP-seq |
| ☒ | ☐ Flow cytometry |
| ☒ | ☐ MRI-based neuroimaging |

