## [Peer Review File · Nature Genetics]

Peer Review Information

Manuscript Title: Yield of genetic association signals from genomes, exomes, and imputation in the UK biobank

Corresponding author name(s): Dr Timothy Thornton, Dr Sheila (M) Gaynor, Dr Aris Baras, Dr Gonçalo (R) Abecasis

Reviewer Comments & Decisions:

Decision Letter, initial version:

31st October 2023

Dear Tim,

Your Article "Yield of genetic association signals from genomes, exomes, and imputation in the UK biobank" has been seen by two referees. You will see from their comments below that, while they find your work of interest, they have raised several relevant points. We are interested in the possibility of publishing your study in Nature Genetics, but we would like to consider your response to these points in the form of a revised manuscript before we make a final decision on publication.

To guide the scope of the revisions, the editors discuss the referee reports in detail within the team, including with the chief editor, with a view to identifying key priorities that should be addressed in revision, and sometimes overruling referee requests that are deemed beyond the scope of the current study. In this case, we ask that you address all technical queries related to the comparative analyses and their interpretation, extending the analyses where feasible as requested by the referees and revising the presentation accordingly. We hope you will find this prioritized set of referee points to be useful when revising your study. Please do not hesitate to get in touch if you would like to discuss these issues further.

We therefore invite you to revise your manuscript taking into account all reviewer and editor comments. Please highlight all changes in the manuscript text file. At this stage, we will need you to upload a copy of the manuscript in MS Word .docx or similar editable format.

*2) If you have not done so already, please begin to revise your manuscript so that it conforms to our Article format instructions, available here. Refer also to any guidelines provided in this letter.

Please be aware of our guidelines on digital image standards.

[redacted]

We hope to receive your revised manuscript within 8-12 weeks. If you cannot send it within this time, please let us know.

Sincerely,
Kyle

Kyle Vogan, PhD

Senior Editor
Nature Genetics
<https://orcid.org/0000-0001-9565-9665>

Referee expertise:

Referee #1: Genetics, complex traits, genome-wide association

Referee #2: Genetics, complex traits, genome-wide association

Reviewers' Comments:

Reviewer #1:
Remarks to the Author:

Thanks for asking me to review this interesting and timely paper from the team at Regeneron. Using data from UK Biobank, it compares the trade-off in discoveries vs cost/effort as they relate to different strategies for gathering large-scale genetic data. Those strategies are array-based genotyping with imputation (here "IMP"), whole exome sequencing ("WES"), and whole genome sequencing ("WGS"), and various combinations thereof, of which "WES+IMP" is the most informative. UK Biobank is, as far as I know, unique in having deployed all three genetic platforms, making for a far less biased analysis than might have been possible using other data sets (where one would have had to downsample the WGS data to generate the WES and IMP equivalents).

The findings are pretty much in line with expectation. IMP doesn't work well for rare variants and is especially poor for gene-level tests that are reliant on very rare alleles. WES is very good for exonic variants (single-variant and gene-based) but poor on non-coding variants (by design!). Because WES and IMP have complementary strengths and gaps, the combination works very well. WGS offers little extra (in these comparisons at least), because the only additional variants it captures that are missed by WES+IMP are rare non-coding variants: on their own, these variants are usually not powered to generate significant associations (especially at the threshold advocated here - of $5E-12$); and, unlike coding variants, we have no effective tools (yet) for combining these variants to perform tests analogous to gene-based aggregate tests. The authors conclude that, given the additional costs of WGS over WES+IMP, the most effective use of a given genetic data generation budget would be to perform WES+IMP.

It's difficult to argue with that conclusion in this setting. And the manuscript serves a very useful purpose by making that case, and providing quantitative data that supports those intuitions.

Having said that, I do think there's a case to be made that the authors are viewing this from a very particular vantage point, and that there are circumstances where this conclusion could be questioned. Some of these they mention in the discussion, but this reviewer would like to see these given more of an airing.

* The most obvious relates to the ancestral background of the samples and the availability of reference sequence that can be used for the imputation: not a problem for UKBB, but there are still

large swathes of humanity for whom the IMP part of the WES+IMP strategy is not currently possible.

* The second relates to the WES+IMP vs WGS comparison not being an entirely fair fight in that the WES approach is tailored by design to enabling gene-based tests, but the absence of analogous approaches for grouping non-coding variants "disables" one of the potential benefits of the WGS approach. Now, we can argue long into the night when (whether) such methods will be possible, and indeed, whether, even if they were, this would ever be as impactful as finding a gene-based aggregate signal. The authors do cover this briefly in the discussion.

* The third relates to value of WGS for other variant types (this is covered in the discussion)

* The fourth relates to the likely narrowing of the cost differential for WES+IMP over WGS. As described here, the WES+IMP approach involves running two different assays, with all the attendant logistical costs of that. Of course, there are methods now that recover the equivalent data from a one-step sequencing assay (though that's not mentioned here), but even then, there are selection steps in the workflow not required for WGS. For how long will it be the case that the differential costs will be that meaningful?

* The fifth concerns the overall costs of a study. A differential cost of say \$100 vs \$300 for WES+IMP vs WGS may look like a massive differential that enables 3x increase in sample size, but in the context of a study where (say) it costs \$1000 to ascertain each sample (so it becomes \$1100 vs \$1300), the impact is much more limited. From that perspective, the added investment in WGS as a single "evergreen" assay that is (mostly) futureproofed, and ready to benefit from advances in non-coding variant aggregation methods, doesn't look quite so profligate.

The paper is well written, and I am confident in the analyses performed by this very capable and experienced team. I do have some (mostly minor) comments on the text. As the authors haven't included line numbers, I hope my approximations on the line numbers per page will help identify the specific pieces of text to which I refer.

* page 3, line 1: please clarify the ancestral composition of these 149K participants

* page 3, line 4: ditto for TopMed reference panel.

* page 4, line 11 (and elsewhere): clarify what is meant by "variants per individual"? Is that "variant alleles per individual"? If so, how defined?

* Figure 2: I was confused about the n-47545 data sets shown here till I read much further into the paper. Some comment in the figure legend would be helpful.

* page 5, line 4: how were these 100 traits selected? I read in the methods later that they were traits that were known to have rare variant associations? Isn't that "tipping the scales" in favor of certain types of analysis?

* page 6, line 1: it would be good to be transparent in the main text about the threshold used here. It seems from methods that the choice was made to adjust the usual genome-wide significance threshold (around $5E-9$ give or take, or maybe $5E-10$ for RV too) for the 100 traits, so that the threshold used here was $5E-12$ for single variant analysis. One could argue that the trait correction is

overkill, but I am OK with this decision given the focus of the paper on relative comparisons. It would be good to be reassured that use of a "per trait" significance threshold (which to me is more logical, and in tune with the way that GWAS data are reported) did not materially alter the conclusions of the paper (I imagine not).

* Similarly, it would be good to state the replication thresholds in the main text.

* page 7, line 15: missing word (presumably "to")

* page 14, line 4: why give this bias towards Hispanic and Latin American ancestral origin, in a sample from the UK, where there are almost no individuals with this ancestral background?

* The examples (in SF7 and 8) of variants that were seen in WGS (but not IMP), and vice versa, don't seem that compelling, given that it's clear from the LZ plots that both were picked up by both WGS and IMP (and significant at single-trait significance levels), and any differential is simply down to the fact that one made it to $5E-12$, and the other didn't.

Reviewer #2:
Remarks to the Author:

In this brief manuscript, the authors compare the relative efficiency of association testing from WGS versus WES versus WES plus imputation from genotyping arrays (WES+IMP) to identify associations in 150k UK Biobank participants for which all three datatypes have been released. For the data studied, they come to the unambiguous conclusion that WES+IMP by far outranks other approaches with regards to association yield, with WGS adding only 1% associations despite a 5-fold higher number of variants assayed. When expanded to the full exome-sequenced UKB cohort, WES+IMP yields a proportionally higher number of associations than expected from increasing sample size alone.

The manuscript is very well written. The analyses are straightforward and appear to be well done, particularly at this scale. The results are not at all surprising: that WGS at current cohort sizes primarily yields variants with insufficient power for association testing, especially when applied to already well studied populations like UKB, is well known. However, this study is the first to systematically conduct such comparative analysis at population scale, for which it uses the to date largest available data. Given the substantial cost benefits of WES+IMP over WGS, this manuscript may thus provide important guidance on the optimal design of future genetic analysis of large cohorts.

1. One of my concerns is that the reasoning of this study is somewhat circular: The power of imputation panels to impute also rarer variants correlates with the number of individuals sequenced. This is evidenced by the improvements the TopMed panel yielded over 1000G and a UKB imputation panel - as e.g. proposed by Halldorsson et al., doi: <https://doi.org/10.1101/2021.11.16.468246> - will likely yield over TopMed. WGS most comprehensively covers the entire allele frequency spectrum across the entire genome and is thus the most suitable substrate for generating imputation panels. WGS at population-scale thus still seems to have its place and will remain necessary in the future, if only to improve (population-specific) imputation accuracy, no?

2. Related to 1, can the authors predict how many more European genomes will still need to be

sequenced to fully saturate an imputation panel for Europeans? It appears that with only 1% more associations for WGS than WES+IMP (based on TopMed) at a population size of 150k, we are just about to meet this point across all variant classes. How many more individuals will we need to sequence to find *all* associations for variants in distinct MAF buckets (<0.01 , <0.001 , <0.0001 , ...)?

3. The authors have called ancestries among UKB participants, but this information was not used to test how well WES+IMP compares to WGS in non-European ancestries. I understand that the non-EUR samples in the UKB 150k overlapping data are extremely small, yet with the comprehensive phenotype information available in UKB, it may suffice to derive interesting learnings already with small numbers of individuals (even if evidence may remain circumstantial). Without, one could hypothesize that the lack of ancestry-specific imputation panels substantially narrows the generalizability of the authors' main conclusion that "WES+IMP trumps WGS" since this might not even be the case for all UKB subcohorts (let alone cohorts other than UKB).

4. I'm surprised that there were $\sim 1/3$ more missense variants found by WES+IMP relative to WGS (126 vs 88k; Suppl. Fig 3). The authors highlight individual loci where the platforms yield inconsistent results in Suppl. Figures, but this discrepancy is not systematically explained. What are the reasons for such substantial difference, and which approach comes closer to the "truth"? Notably, according to Suppl Table 3, this discrepancy seems not to come from singletons, but primarily variants in the low to medium allele frequency spectrum where the yield of new associations from sequencing data is potentially the highest. Can the authors exclude that part of the difference between WES+IMP and WGS is just artifactual and explained simply by technical differences that lead WGS algorithms to just call fewer of the most informative variants?

5. UKB was sequenced at an average read-depth of 23.5X. The cost argument why readers should choose WES+IMP over WGS to genetically profile their respective population cohort could break away if low-coverage WGS would be performed instead deep sequencing, and many investors are wondering whether they should place a bet on lcWGS relative to WES or deep WGS. Have the authors considered to downsample UKB genomes to just ~ 2 -4X coverage and test how such theoretical lcWGS+IMP (e.g. following the method of Rubinacci et al., Nature Genetics 2023, 55:1088) compares to WES+IMP and WGS? This could provide helpful guidance.

6. The authors also may want to better highlight their finding that, due to the ability of WGS to call substantially more indels and frameshift variants than WES+IMP, the effect sizes of some WGS associations can be expected to be high above those of other platforms (Suppl. Table 3, Suppl. Figs. 10-11). LoF associations are typically easier to follow up experimentally, which would be an argument in favor of WGS, even if the overall numbers of associations are smaller (Halldorsson et al. provide some insightful examples).

7. Halldorsson et al. (doi: <https://doi.org/10.1101/2021.11.16.468246>) introduced a UKB-based reference panel that was claimed to successfully impute rare variants even to very low frequencies and to reliably work also for individuals with non-European ancestry. I am not sure whether the full results of that study have already been released and could be utilized to further refine results from this study, but it would be highly interesting to see by how much this new panel might further improve (or not) association testing relative to the TopMed panel utilized here (please disregard this comment if the UKB panel should be unobtainable at present).

8. Kurki et al. (<https://doi.org/10.1038/s41586-022-05473-8>) and other papers on the Finnish

population isolate have convincingly demonstrated that, with a population-specific imputation panel that is based on a modest number of genomes with the same haplotype structure, imputation to very low allele frequencies is possible from array-based genotypes alone (= "small-scale WGS+IMP"). Such scenarios may make population-level WES obsolete in some populations and should at least be discussed as an additional alternative.

Other:

9. typo on p.2: IFNRA2 -> IFNAR2

10. Figure 1a, non-coding abbreviated with both "N" and "NC" -> please choose

11. Figure 1d legend, please explain abbreviation "AAC"

Author Rebuttal to Initial comments

Response to Reviewers' Comments

Reviewer #1

Thanks for asking me to review this interesting and timely paper from the team at Regeneron. Using data from UK Biobank, it compares the trade-off in discoveries vs cost/effort as they relate to different strategies for gathering large-scale genetic data. Those strategies are array-based genotyping with imputation (here "IMP"), whole exome sequencing ("WES"), and whole genome sequencing ("WGS"), and various combinations thereof, of which "WES+IMP" is the most informative. UK Biobank is, as far as I know, unique in having deployed all three genetic platforms, making for a far less biased analysis than might have been possible using other data sets (where one would have had to downsample the WGS data to generate the WES and IMP equivalents).

Response: We thank the reviewer for these comments.

The findings are pretty much in line with expectation. IMP doesn't work well for rare variants and is especially poor for gene-level tests that are reliant on very rare alleles. WES is very good for exonic variants (single-variant and gene-based) but poor on non-coding variants (by design!). Because WES and IMP have complementary strengths and gaps, the combination works very well. WGS offers little extra (in these comparisons at least), because the only additional variants it captures that are missed by WES+IMP are rare non-coding variants: on their own, these variants are usually not powered to generate significant associations (especially at the threshold advocated here - of $5E-12$); and, unlike coding variants, we have no effective tools (yet) for combining these variants to perform tests analogous to gene-based aggregate tests. The authors conclude that, given the additional costs of WGS over WES+IMP, the most effective use of a given genetic data generation budget would be to perform WES+IMP.

It's difficult to argue with that conclusion in this setting. And the manuscript serves a very useful purpose by making that case, and providing quantitative data that supports those intuitions.

Response: We agree and thank the reviewer for their perspective and summary of our findings.

Having said that, I do think there's a case to be made that the authors are viewing this from a very particular vantage point, and that there are circumstances where this conclusion could be questioned. Some of these they mention in the discussion, but this reviewer would like to see these given more of an airing.

Response: We appreciate the constructive feedback. As summarized below, we tried to address all reviewer suggestions in the revised manuscript.

* The most obvious relates to the ancestral background of the samples and the availability of reference sequence that can be used for the imputation: not a problem for UKBB, but there are still large swathes of humanity for whom the IMP part of the WES+IMP strategy is not currently possible.

Response: The reviewer raises an important point that accurate imputation may not be possible for some populations around the world who are not adequately represented in existing WGS reference panels. We have added to the Discussion the following text on imputation, highlighting the need for more genetic studies and WGS reference panels that are representative of the diverse populations around the world:

“The IMP component of the WES+IMP approach requires a suitable WGS reference panel. There are still many populations that are insufficiently represented in contemporary reference panels, and there remains a need for diverse WGS reference datasets to increase imputation accuracy and genetic discovery across all ancestries.”

We also now include additional context for IMP analysis as well as an analysis of imputation accuracy as detailed in our response to comment 3 from Reviewer 2.

* The second relates to the WES+IMP vs WGS comparison not being an entirely fair fight in that the WES approach is tailored by design to enabling gene-based tests, but the absence of analogous approaches for grouping non-coding variants "disables" one of the potential benefits of the WGS approach. Now, we can argue long into the night when (whether) such methods will be possible, and indeed, whether, even if they were, this would ever be as impactful as finding a gene-based aggregate signal. The authors do cover this briefly in the discussion.

Response: The reviewer highlights an important challenge in contemporary analysis of non-coding variants. Approaches for grouping non-coding variants remain in development and thus we cannot conclusively state their future value. We have evaluated current methods using sliding windows and enhancer-based groupings. Thus far, these analyses did not provide additional discovery beyond the single variant and gene-based association results. We have included in the Discussion the following revised text: “In the future, if such aggregate approaches can successfully group rare non-coding variants, this could provide a potential benefit of WGS.”

* The third relates to value of WGS for other variant types (this is covered in the discussion)

Response: We agree.

* The fourth relates to the likely narrowing of the cost differential for WES+IMP over WGS. As described here, the WES+IMP approach involves running two different assays, with all the attendant logistical costs of that. Of course, there are methods now that recover the equivalent data from a one-step sequencing assay (though that's not mentioned here), but even then, there are selection steps in the workflow not required for WGS. For how long will it be the case that the differential costs will be that meaningful?

Response: The reviewer raises an important point that these conclusions relate to the current setting of common assay approaches and costs. As both of these continue to evolve, it would be prudent to re-evaluate with the latest technologies and cost basis. We have expanded the discussion to state that

“The cost and complexity of biobank-scale WGS can be substantial compared to WES+IMP, yet our analysis indicated that the two approaches have comparable association yield (1% higher yield for single variants and 1% lower yield for gene-based tests with WGS). We note that approaches for WGS and WES+IMP all continue to evolve. For instance, labs at the Regeneron Genetics Center use a combined exome and common variant capture assay, eliminating the need for a separate genotyping array. Others have used low coverage whole genome sequencing in combination with WES. And, still others, continue to develop long-read alternatives to extend the potential of WGS. Optimal use of resources must also consider the value of generating genetic data on additional samples or obtaining deeper phenotypic characterization (perhaps proteomics and RNA-sequencing) on the same samples as alternatives to broad WGS [41-44].”

* The fifth concerns the overall costs of a study. A differential cost of say \$100 vs \$300 for WES+IMP vs WGS may look like a massive differential that enables 3x increase in sample size, but in the context of a study where (say) it costs \$1000 to ascertain each sample (so it becomes

\$1100 vs \$1300), the impact is much more limited. From that perspective, the added investment in WGS as a single "evergreen" assay that is (mostly) futureproofed, and ready to benefit from advances in non-coding variant aggregation methods, doesn't look quite so profligate.

Response: There are multiple contributing factors to the cost of a study, necessitating the evaluation of all aspects when designing and implementing a study. We have included additional points, included in the expanded discussion above, to seek to highlight this. Specifically, we sought to address this by noting the full context is important for evaluating approaches and cost, and that WGS continues to undergo development to provide as comprehensive an approach as possible. We added an additional sentence "Selecting the best strategy for each study should also take into consideration study design and goals, considering for instance the cost of recruitment (which is often relatively modest for biobank studies, but very high for studies of rare Mendelian disorders) relative to genome assay and analysis costs." to address this further.

The paper is well written, and I am confident in the analyses performed by this very capable and experienced team. I do have some (mostly minor) comments on the text. As the authors haven't included line numbers, I hope my approximations on the line numbers per page will help identify the specific pieces of text to which I refer.

Response: We are grateful for these comments and apologize for the oversight of missing line numbers. We have added line numbers to the revised manuscript.

* page 3, line 1: please clarify the ancestral composition of these 149K participants

Response: We have added the following description to the text, summarizing Supplementary Table 1: "Most of this analytical sample was of European ancestry (95%), with the remaining individuals having African (2%), South Asian (2%), and other ancestries (<1%)."

* page 3, line 4: ditto for TopMed reference panel.

Response: We have expanded the description of the TOPMed reference panel with the following text: "We extended this data to an additional 111,333,957 variants through imputation using TOPMed Freeze 8 genomes as a reference panel, which is an ancestrally diverse imputation panel comprised of genomes of European, African, Hispanic/Latino, Asian, and Other/Multiple ancestries (see Supplementary Figure 12 for an evaluation of imputation performance by ancestry) [22, 30, 31]."

* page 4, line 11 (and elsewhere): clarify what is meant by "variants per individual"? Is that "variant alleles per individual"? If so, how defined?

Response: Variants per individual refers to variant alleles per individual, defined as the number of alternate alleles observed in each individual. We have updated the text and use the phrase “variant alleles per individual” in all places to improve clarity.

* Figure 2: I was confused about the n-47545 data sets shown here till I read much further into the paper. Some comment in the figure legend would be helpful.

Response: We apologize for the confusion and have added a sentence to the figure legend to improve clarity on the different data sets analyzed: “Summarized results are also provided for the full UKB sample with WES+IMP data (n=468,169) and a subset of the UKB sample (n=47,545), downsampled to also have a 1:3 ratio with the primary analytical sample used for the majority of our analyzes (n=149,195).”

* page 5, line 4: how were these 100 traits selected? I read in the methods later that they were traits that were known to have rare variant associations? Isn't that "tipping the scales" in favor of certain types of analysis?

Response: Our goal for trait selection was to include traits for evaluation that were adequately powered for genetic discovery analysis and that also had significant association signals. The 100 traits, now provided as Supplementary Table 10, were selected from among the 492 traits analyzed by Backman *et al* (2021) that had rare variant associations. Traits were pruned from this set to eliminate trait redundancies, have sufficient case counts for analysis of binary traits in our main analytical sample of 149K (Backman *et al* analyzed 454K), and prioritized higher genetic heritability for quantitative traits. We retained 80 quantitative and 20 binary traits.

This selection approach may result in a small bias against IMP and in favor of the sequencing-based analyses (WES and WGS) for detecting the rare variant association signals, with the expectation that high coverage WGS should have broad capture including all regions sequenced by WES and subsequently identified in Backman *et al*. However, we believe this bias against IMP is mitigated by the approach of combining IMP with WES (WES+IMP). We have edited the Methods section and now provide the entire trait list in the supplementary materials: “A set of 100 traits (listed in Supplementary Table 10) were selected to permit generalizable conclusions on genetic association discovery yield for quantitative and binary traits.”

* page 6, line 1: it would be good to be transparent in the main text about the threshold used here. It seems from methods that the choice was made to adjust the usual genome-wide significance threshold (around 5E-9 give or take, or maybe 5E-10 for RV too) for the 100 traits, so that the threshold used here was 5E-12 for single variant analysis. One could argue that the trait correction is overkill, but I am OK with this decision given the focus of the paper on relative comparisons. It would be good to be reassured that use of a "per trait" significance threshold

(which to me is more logical, and in tune with the way that GWAS data are reported) did not materially alter the conclusions of the paper (I imagine not).

Response: We used a stringent threshold to account for the large multiple testing burden resulting from the large number of variants and traits being analyzed. However, as we now show in Supplementary Table 9 (which summarizes results using relaxed significance thresholds), this does not change our conclusions.

We have added the following text to the Single Variant Tests section: “We used a significance threshold of $P=5 \times 10^{-12}$ for the main analysis; a threshold of $P=5 \times 10^{-11}$ was used for classifying a signal as shared across platforms when at least one platform reached $P=5 \times 10^{-12}$.” We have revised the Gene-Based Tests description to include the sentence “We summarized all association tests for gene-trait pairs using GENE_P [33], a single, unified gene-based p-value where a significance and replication threshold of $P=2.6 \times 10^{-8}$ was used (Methods). We found the overall conclusions from the results to be similar when examining the individual gene-based tests, using different allele frequency thresholds, or applying a less conservative significance threshold (Supplementary Table 9).”

We also have added to the supplement the results from the additional experiments when using a less conservative threshold for significance. We thus summarized the results of our single variant analysis using a significance threshold of $P=5 \times 10^{-10}$ and the results of our gene-based analysis using a significant threshold of $P=2.6 \times 10^{-6}$. Supplementary Table 9 is given below:

	Single variant analysis		Gene-based analysis	
	Number of signals, main analysis	Number of signals, relaxed threshold	Number of signals, main analysis	Number of signals, relaxed threshold
Shared	3,470	4,889	528	859
WGS only	64	241	10	21
WES+IMP only	36	167	18	38

These results show that under this adjusted threshold, again the majority of signals in both single variant and gene-based results are identified by both approaches. We observe that slightly more signals are identified only by the WES+IMP approach under this reduced threshold, but our conclusions remain unchanged. We now describe this in the Single Variant Tests section with the following text: “We also performed the association analysis described above using a less conservative significance threshold of $P=5 \times 10^{-10}$ which resulted in more genome-wide significant signals, as expected, but did not change overall conclusions on association yield for the different approaches (Supplementary Table 9).”

* Similarly, it would be good to state the replication thresholds in the main text.

Response: We have now included the replication thresholds in the revised manuscript as described in the previous response.

* page 7, line 15: missing word (presumably "to")

Response: We have revised the text to say: "When we extended the WES+IMP analysis from 149,195 individuals to 468,169 individuals (a three-fold increase in sample size), the number of single variant signals increased more than four-fold from 3,506 to 15,565."

* page 14, line 4: why give this bias towards Hispanic and Latin American ancestral origin, in a sample from the UK, where there are almost no individuals with this ancestral background?

Response: This general ancestry assignment approach was taken to capture the major continental ancestry groups represented in the reference panel (HapMap3). We acknowledge this approach aligns more with US studies but note that it has not materially impacted the assignment in this analysis as <100 individuals were classified as "Hispanic and Latin American ancestral origin".

* The examples (in SF7 and 8) of variants that were seen in WGS (but not IMP), and vice versa, don't seem that compelling, given that it's clear from the LZ plots that both were picked up by both WGS and IMP (and significant at single-trait significance levels), and any differential is simply down to the fact that one made it to 5E-12, and the other didn't.

Response: We have updated these figures and included new examples to demonstrate more clearly settings of platform-significant findings where the conclusion of significance is not influenced by significance threshold.

Reviewer #2

In this brief manuscript, the authors compare the relative efficiency of association testing from WGS versus WES plus imputation from genotyping arrays (WES+IMP) to identify associations in 150k UK Biobank participants for which all three datatypes have been released. For the data studied, they come to the unambiguous conclusion that WES+IMP by far outranks other approaches with regards to association yield, with WGS adding only 1% associations despite a 5-fold higher number of variants assayed. When expanded to the full exome-sequenced UKB cohort, WES+IMP yields a proportionally higher number of associations than expected from increasing sample size alone.

The manuscript is very well written. The analyses are straightforward and appear to be well done, particularly at this scale. The results are not at all surprising: that WGS at current cohort sizes primarily yields variants with insufficient power for association testing, especially when applied to already well studied populations like UKB, is well known. However, this study is the

first to systematically conduct such comparative analysis at population scale, for which it uses the to date largest available data. Given the substantial cost benefits of WES+IMP over WGS, this manuscript may thus provide important guidance on the optimal design of future genetic analysis of large cohorts.

Response: We thank the reviewer for the positive feedback on the manuscript.

1. One of my concerns is that the reasoning of this study is somewhat circular: The power of imputation panels to impute also rarer variants correlates with the number of individuals sequenced. This is evidenced by the improvements the TopMed panel yielded over 1000G and a UKB imputation panel - as e.g. proposed by Halldorsson et al., doi: <https://doi.org/10.1101/2021.11.16.468246> – will likely yield over TopMed. WGS most comprehensively covers the entire allele frequency spectrum across the entire genome and is thus the most suitable substrate for generating imputation panels. WGS at population-scale thus still seems to have its place and will remain necessary in the future, if only to improve (population-specific) imputation accuracy, no?

Response: We agree and previously included text in the manuscript that WGS reference panels are important to enable quality imputation in diverse samples. Our analyses further demonstrate that imputation-based analyses can be highly informative for genetic association discovery, and when combined with WES, provide comparable genetic association discovery to WGS with respect to single variant and gene-based association analyses.

We have now expanded the Discussion section on imputation, and include the following text emphasizing the continued need for WGS reference sets:

- “There are still many populations that are insufficiently represented in contemporary reference panels, and there remains a need for diverse WGS reference datasets to increase imputation accuracy and genetic discovery across all ancestries.”
- “WGS studies that enable improved imputation reference panels [13, 21, 22, 40] can thus enhance genetic association discovery for future studies.”

2. Related to 1, can the authors predict how many more European genomes will still need to be sequenced to fully saturate an imputation panel for Europeans? It appears that with only 1% more associations for WGS than WES+IMP (based on TopMed) at a population size of 150k, we are just about to meet this point across all variant classes. How many more individuals will we need to sequence to find *all* associations for variants in distinct MAF buckets (<0.01, <0.001, <0.0001, ...)?

Response: The practical saturation point for imputation and association analyses remains an active research area but we speculate that it might correspond to a well-matched reference panel of 3-10% of the sample used for association analysis. With a reference panel larger than this size, newly imputed very rare variants are usually not well powered for association analyses. Saturating association analyses, whether with sequencing or imputation, is more challenging. Current evidence suggests that the number of detectable association signals is still rising quickly as sample size increases, across all frequencies.

3. The authors have called ancestries among UKB participants, but this information was not used to test how well WES+IMP compares to WGS in non-European ancestries. I understand that the non-EUR samples in the UKB 150k overlapping data are extremely small, yet with the comprehensive phenotype information available in UKB, it may suffice to derive interesting learnings already with small numbers of individuals (even if evidence may remain circumstantial). Without, one could hypothesize that the lack of ancestry-specific imputation panels substantially narrows the generalizability of the authors' main conclusion that "WES+IMP trumps WGS" since this might not even be the case for all UKB subcohorts (let alone cohorts other than UKB).

Response: The importance of and need for high-quality imputation across different ancestry groups was also importantly raised by reviewer 1. Our UKB sample is primarily of European ancestry, with a small subset of non-European ancestry individuals. We now include details on the ancestries of the study subjects in the results section: "Most of this analytical sample was of European ancestry (95%), with the remaining individuals having African (2%), South Asian (2%), and other ancestries (<1%). The UKB used a custom designed SNP array with 805,426 variants designed to capture common variation and selected high-value protein altering variants. We extended this data to an additional 111,333,957 variants through imputation using TOPMed Freeze 8 genomes as a reference panel, which is an ancestrally diverse imputation panel comprised of genomes of European, African, Hispanic/Latino, Asian, and Other/Multiple ancestries (see Supplementary Figure 12 for an evaluation of imputation performance by ancestry) [22, 30, 31]."

We also conducted an analysis evaluating imputation accuracy by ancestry in UKB using the whole genome sequencing data and genotypes imputed from the TOPMed reference panel. We binned variants by frequency and calculated the imputation accuracy (R^2) for each ancestry group, as shown below and included as Supplementary Figure 12.

Imputation accuracy was highest for European ancestry individuals followed by African ancestry individuals in UKB, which is expected since these two ancestry groups have the highest representation in TOPMed (40% European ancestry and 29% African ancestry). The lowest imputation accuracy was observed for the East and South Asian individuals in UKB which is also as expected due to the low representation of Asian ancestry populations in TOPMed.

We have added to the Discussion section commentary on imputation performance and our assessment of imputation accuracy across ancestries with the TOPMed imputation reference panel in UKB: “In our IMP analysis with the TOPMed imputation panel, we found imputation accuracy varied by ancestry, with the highest accuracy for European and African ancestry individuals; in contrast, East Asian and South Asian ancestries had the lowest imputation accuracy (Supplementary Figure 12). Alternative panels with improved performance in these ancestries and across the UK have been proposed (Supplementary Figure 13 and [39]). WGS studies that enable improved imputation reference panels [13, 21, 22, 40] can thus enhance genetic association discovery for future studies.”

4. I'm surprised that there were ~1/3 more missense variants found by WES+IMP relative to WGS (126 vs 88k; Suppl. Fig 3). The authors highlight individual loci were the platforms yield inconsistent results in Suppl. Figures, but this discrepancy is not systematically explained. What

are the reasons for such substantial difference, and which approach comes closer to the “truth”? Notably, according to Suppl Table 3, this discrepancy seems not to come from singletons, but primarily variants in the low to medium allele frequency spectrum where the yield of new associations from sequencing data is potentially the highest. Can the authors exclude that part of the difference between WES+IMP and WGS is just artifactual and explained simply by technical differences that lead WGS algorithms to just call fewer of the most informative variants?

Response: We apologize for the confusion and appreciate the opportunity to clarify the findings. Overall, the total number of missense variants found is very similar for WES+IMP and WGS (4,225,468 vs 4,263,339, with most found by both platforms). The reviewer is focusing on platform unique variants, which are a very small fraction of the total. The variants unique to each platform span all frequency thresholds, and there is an enrichment of unique variants observed for the rarest frequencies for WGS. This is reflected in the majority of identified variants being singletons and doubletons where there are the largest counts in Supplementary Table 3 of “WGS only” variants.

We have emphasized the text to note that when we consider the total number of variants detected, the differences are very small. The differences in discovery among platforms result from variant callers, read length, sequence coverage, and paired-read insert sizes – among others. In Supplementary Table 5, we provide the consequences of all signals by approach, clearly showing that nearly all missense variant associations are identified by both approaches. Our conclusion remains unchanged that the individual association signals (most pointing to the exact same variant) and overall association yield (with an ~1% change in the total number of signals) is fairly equivalent for discovery using WGS or WES+IMP.

5. UKB was sequenced at an average read-depth of 23.5X. The cost argument why readers should choose WES+IMP over WGS to genetically profile their respective population cohort could break away if low-coverage WGS would be performed instead deep sequencing, and many investors are wondering whether they should place a bet on lcWGS relative to WES or deep WGS. Have the authors considered to downsample UKB genomes to just ~2-4X coverage and test how such theoretical lcWGS+IMP (e.g. following the method of Rubinacci et al., Nature Genetics 2023, 55:1088) compares to WES+IMP and WGS? This could provide helpful guidance.

Response: Sequencing costs differ across coverage levels, and previous studies have evaluated the implications of implementing lcWGS as a cost-effective alternative to higher coverages, with optimal depth when considering costs for rare variant sequencing studies estimated to be around 15X (Rashkin 2017); more common variants require less depth. We have added a sentence to the discussion to mention the potential of lcWGS as an alternative: “Others have used low coverage whole genome sequencing in combination with WES.”

In our analysis, we saw the differences in yield of genetic association signals for single and gene-based settings were less than 1% when comparing high coverage sequencing to WES+IMP. This suggests that there is little to be gained by lcWGS, as higher WGS coverage will outperform lcWGS and did not demonstrably outperform WES+IMP for genetic association discovery. Also, rare variant studies require sufficient depth to capture more variants in each individual; therefore, lcWGS will not be able to capture the rarest variants such as singletons and doubletons, which comprise 49% and 15% of WES. WES thus provides a key advantage over lcWGS in coding regions and therefore discovery with gene-based association analyses.

6. The authors also may want to better highlight their finding that, due to the ability of WGS to call substantially more indels and frameshift variants than WES+IMP, the effect sizes of some WGS associations can be expected to be high above those of other platforms (Suppl. Table 3, Suppl. Figs. 10-11). LoF associations are typically easier to follow up experimentally, which would be an argument in favor of WGS, even if the overall numbers of associations are smaller (Halldorsson et al. provide some insightful examples).

Response: This is an important point raised by the reviewer. While we did observe more indels and frameshift variants in WGS than WES+IMP, our single variant analyses found the effect sizes between the two approaches did not differ statistically, as evidenced by the overlapping boxplots in Supplementary Figure 10. Most single variant signals were shared and amongst shared signals 96% of these pointed to the exact same variant. For gene-based testing, Supplementary Figure 11 shows lower p-values in WGS than IMP alone. However, the correlation between WGS and WES p-values was 0.98, thus for gene-based testing, the results of the combination of WES+IMP led to results comparable to WGS.

7. Halldorsson et al. (doi: <https://doi.org/10.1101/2021.11.16.468246>) introduced a UKB-based reference panel that was claimed to successfully impute rare variants even to very low frequencies and to reliably work also for individuals with non-European ancestry. I am not sure whether the full results of that study have already been released and could be utilized to further refine results from this study, but it would be highly interesting to see by how much this new panel might further improve (or not) association testing relative to the TopMed panel utilized here (please disregard this comment if the UKB panel should be unobtainable at present).

Response:

Choice of reference panel can impact imputation accuracy and discovery with IMP. Since the difference in association yield in UKB with WGS and WES+IMP is less than 1% when using the TOPMed imputation panel, we do not expect an alternate imputation reference panel to substantively change the overall conclusion of our study. To assess the potential gain in imputation accuracy when using a cohort-specific reference panel, we conducted a data-driven evaluation of imputation accuracy (R^2) between genomes imputed with the TOPMed imputation panel and a UKB-based reference panel generated from 200K UKB phased WGS samples.

Using a sample of 1,000 White British individuals with WGS and array data, imputation accuracy was performed by comparing the imputed data and sequenced calls.

We observe that the UKB-based imputation performs better than TOPMed imputed calls for rare and low frequency variants, as shown in the figure below, where imputation accuracy is provided across bins of minor allele counts.

These results illustrate that a UKB-based imputation panel would improve imputation accuracy over TOPMed, for individuals with White British ancestry as in UKB. We added the figure to the supplementary material (Supplementary Figure 13.) and included the following text on imputation accuracy for the TOPMed and UKB panels in the Discussion: “Alternative panels with improved performance in these ancestries and across the UK have been proposed (Supplementary Figure 13 and [39]).”

8. Kurki et al. (<https://doi.org/10.1038/s41586-022-05473-8>) and other papers on the Finnish population isolate have convincingly demonstrated that, with a population-specific imputation panel that is based on a modest number of genomes with the same haplotype structure, imputation to very low allele frequencies is possible from array-based genotypes alone (= “small-scale WGS+IMP”). Such scenarios may make population-level WES obsolete in some populations and should at least be discussed as an additional alternative.

Response: We have included this reference in the revised manuscript and added a sentence to note this observation in the discussion: “Some have argued that IMP alone may be sufficient for genetic association discovery, particularly in population isolates [38]. We found sequencing-based approaches enable discoveries at frequencies below those accessible through imputation, increasing gene-based findings by 30%, so that a combination of WES+IMP was most effective for UKB.”

9. typo on p.2: IFNRA2 -> IFNAR2

Response: The typo has been revised.

10. Figure 1a, non-coding abbreviated with both “N” and “NC” -> please choose

Response: The figure has been revised to exclusively use “NC”.

11. Figure 1d legend, please explain abbreviation “AAC”

Response: The figure legend has been revised to state that “*The number of non-coding and coding variants observed at the lowest alternative allele counts (AAC) for the WES+IMP and WGS data.*”

Decision Letter, first revision:

16th January 2024

Dear Tim,

Your revised manuscript "Yield of genetic association signals from genomes, exomes, and imputation in the UK biobank" (NG-A63344R) has been seen by the original referees. As you will see from their comments below, they are satisfied with the changes made in response to their previous comments, and therefore we will be happy in principle to publish your study in Nature Genetics as an Article pending final revisions to satisfy their remaining requests and to comply with our editorial and formatting guidelines.

We are now performing detailed checks on your paper, and we will send you a checklist detailing our editorial and formatting requirements soon. Please do not upload the final materials or make any revisions until you receive this additional information from us.

Thank you again for your interest in Nature Genetics. Please do not hesitate to contact me if you have any questions.

Sincerely,

Kyle

Kyle Vogan, PhD
Senior Editor
Nature Genetics
<https://orcid.org/0000-0001-9565-9665>

Reviewer #1 (Remarks to the Author):

The authors have made sensible and appropriate revisions to address the points raised by the reviewers. No further comments.

Reviewer #2 (Remarks to the Author):

My comments have been well addressed. I congratulate the authors to a very nice study. My few remaining suggestions are very minor and should be addressable without another round of review.

1. To avoid confusion I suggest that the sentence introduced in the first paragraph of Discussion in response to my Comment 8 should be modified as such (or similar): "Some have argued that IMP from a cohort-specific reference panel alone may be sufficient for genetic association discovery, particularly in population isolates [38]. In UKB, we found sequencing-based approaches enable discoveries at frequencies below those accessible through imputation..."

2. Typo: Please remove surplus "that" in final sentence of Discussion ("Efficient genetic association analyses...")

3. Typo: "individuals" in Legend to new Suppl. Figure 13

Heiko Runz

Author Rebuttal, first revision:

Reviewer #1

The authors have made sensible and appropriate revisions to address the points raised by the reviewers. No further comments.

Response: We thank the reviewer for all feedback and improving the manuscript.

Reviewer #2

My comments have been well addressed. I congratulate the authors to a very nice study. My few remaining suggestions are very minor and should be addressable without another round of review.

Response: We appreciate the reviewer's comments and thorough review of the manuscript.

1. To avoid confusion I suggest that the sentence introduced in the first paragraph of Discussion in response to my Comment 8 should be modified as such (or similar): "Some have argued that IMP from a cohort-specific reference panel alone may be sufficient for genetic association discovery, particularly in population isolates [38]. In UKB, we found sequencing-based approaches enable discoveries at frequencies below those accessible through imputation..."

Response: We have edited the discussion to improve clarity, which now reads: "Some have argued that IMP from a well-matched reference panel may be sufficient for genetic association discovery, particularly in population isolates [38]. In UKB, we found sequencing-based approaches enable discoveries at frequencies below those accessible through imputation..."

2. Typo: Please remove surplus "that" in final sentence of Discussion ("Efficient genetic association analyses...")

Response: We have removed the surplus "that" from the final sentence.

3. Typo: "individuals" in Legend to new Suppl. Figure 13

Response: We have adjusted the typo "individual."

Final Decision Letter:

23rd August 2024

Dear Tim,

I am delighted to say that your manuscript "Yield of genetic association signals from genomes, exomes, and imputation in the UK biobank" has been accepted for publication in an upcoming issue of Nature Genetics.

Your paper will be published online after we receive your corrections and will appear in print in the next available issue. You can find out your date of online publication by contacting the Nature Press Office (press@nature.com) after sending your e-proof corrections.

Before your paper is published online, we will be distributing a press release to news organizations worldwide, which may very well include details of your work. We are happy for your institution or funding agency to prepare its own press release, but it must mention the embargo date and Nature Genetics. Our Press Office may contact you closer to the time of publication, but if you or your Press Office have any enquiries in the meantime, please contact press@nature.com.

Please note that Nature Genetics is a Transformative Journal (TJ). Authors may publish their research with us through the traditional subscription access route or make their paper immediately open access through payment of an article-processing charge (APC). Authors will not be required to make a final decision about access to their article until it has been accepted. Find out more about Transformative Journals

Authors may need to take specific actions to achieve compliance with funder and institutional open access mandates. If your research is supported by a funder that requires immediate open access (e.g. according to Plan S principles), then you should select the gold OA route, and we will direct you to the compliant route where possible. For authors selecting the subscription publication route, the journal's standard licensing terms will need to be accepted, including [a href="https://www.nature.com/nature-portfolio/editorial-policies/self-archiving-and-license-to-publish"](https://www.nature.com/nature-portfolio/editorial-policies/self-archiving-and-license-to-publish). Those licensing terms will supersede any other terms that the author or any third party may assert apply to any version of the manuscript.

If you have posted a preprint on any preprint server, please ensure that the preprint details are

updated with a publication reference, including the DOI and a URL to the published version of the article on the journal website.

If you have not already done so, we strongly recommend that you upload the step-by-step protocols used in this manuscript to protocols.io. protocols.io is an open online resource that allows researchers to share their detailed experimental know-how. All uploaded protocols are made freely available and are assigned DOIs for ease of citation. Protocols can be linked to any publications in which they are used and will be linked to from your article. You can also establish a dedicated workspace to collect all your lab Protocols. By uploading your Protocols to protocols.io, you are enabling researchers to more readily reproduce or adapt the methodology you use, as well as increasing the visibility of your protocols and papers. Upload your Protocols at <https://protocols.io>. Further information can be found at <https://www.protocols.io/help/publish-articles>.

Sincerely,
Kyle

Kyle Vogan, PhD
Senior Editor
Nature Genetics
<https://orcid.org/0000-0001-9565-9665>